# The influence of random number generation in dissipative particle dynamics simulations using a cryptographic hash function

**Kiyoshiro Okada, Paul E. Brumby**◯**, Kenji Yasuoka\***

Department of Mechanical Engineering, Keio University, Yokohama, Kanagawa, Japan

\* yasuoka@mech.keio.ac.jp

## Abstract

The tiny encryption algorithm (TEA) is widely used when performing dissipative particle dynamics (DPD) calculations in parallel, usually on distributed memory systems. In this research, we reduced the computational cost of the TEA hash function and investigated the influence of the quality of the random numbers generated on the results of DPD calculations. It has already been established that the randomness, or quality, of the random numbers depend on the number of processes from internal functions such as SHIFT, XOR and ADD, which are commonly referred to as "rounds". Surprisingly, if we choose seed numbers from high entropy sources, with a minimum number of rounds, the quality of the random numbers generated is sufficient to successfully perform accurate DPD simulations. Although it is well known that using a minimal number of rounds is insufficient for generating high-quality random numbers, the combination of selecting good seed numbers and the robustness of DPD simulations means that we can reduce the random number generation cost without reducing the accuracy of the simulation results.

**Data Availability Statement:** All relevant data are within the manuscript and its Supporting information files.

## Introduction

Particle-based simulation methods are powerful tools with which to study microscopic stochastic systems. In order to make valid comparisons with experimental data, large-scale simulations are often required. Typically, this entails the simulation of large numbers of particles. Although this can be costly in terms of computational resources, the rapid development of faster and more efficient hardware such as many-core computers or graphic processing units (GPUs), coupled with new and improved calculation methods, permits us to perform such large-scale simulations. For Example, Ayuba et al. simulated systems composed of more than 100 million particles [1] using a many-core computer and an application called the Framework for Developing Particle Simulator (FDPS) [2], which allows performance optimisation of particle-based simulations on such hardware.

The use of coarse-grained (CG) methods is also an effective way to perform large-scale simulations. One such method is dissipative particle dynamics (DPD) [3–5]. Due to the broad applicability of this method, it has seen use for the simulation of polymers, [6–10] biological

**Funding:** The authors received no specific funding for this work.

**Competing interests:** The authors have declared that no competing interests exist.

membranes, [11–14] colloids, [15–18] and other complex materials [19–22]. The future prospects and potential of DPD were discussed recently by Español and Warren [23].

A parallelised version of the DPD algorithm has been incorporated into a publicly available simulation code [24–26]. The result of this is that computationally efficient DPD simulations are now much more accessible to researchers. On the other hand, generating random numbers for parallel calculations is a non-trivial problem and has often been the topic of discussions in this area [27–31]. What makes this particularly difficult for DPD calculations is that the same pair-wise random forces are required for each pair of interacting particles. This presents a problem when particles may be allocated to different computational nodes. Since the random number used should be the same for both nodes, communication between nodes is required and this can cause a significant decrease in computational efficiency. To address this issue, Phillips et al. [32] employed a hash-based random number generator using the tiny encryption algorithm (TEA) [33] for their DPD calculations. Further to this, the effects of different PRNGs including TEA for parallel DPD simulations were considered by Nguyen and Plimpton [34]. Based on these ideas, more efficient random number generation methods have subsequently been developed [24, 35]. Here, we focus on the use of the TEA method since it is easier to adjust the calculation speed and the quality of the random numbers generated. The properties of these random numbers and the calculation cost of the TEA was evaluated by Zafar et al. [31] and they suggested that the number of rounds, which is the number of repetitions of the round function in the TEA, should be at least 8. This recommendation was made with consideration of the balance between calculation cost and the randomness of the generated random numbers, which we shall refer to here as their quality.

We question if it is indeed necessary to go to such lengths to ensure that the random numbers used in DPD simulations are of sufficiently high quality. We also suggest an easier and more efficient source for TEA seeds, which are usually taken from the number of steps, the particle ID, or the first 11 bits of the mantissa of the three-dimensional particle velocity components [24]. In this paper, we use the relative distance between particle pairs as seed numbers and investigate the usage of the TEA with fewer than 8 rounds. For DPD simulations performed with the TEA, we looked at how the quality of the randomly generated numbers changes as the number of rounds decreases. This was done by checking only the uniformity and correlations in successive steps. By measuring these two properties we were able to evaluate the randomness, or quality, of the random numbers produced. Further to this, we also applied the NIST test sets to provide additional validation to our findings.

We also studied, in detail, how the number of rounds effects several of the physical properties one may measure from systems simulated using DPD. As a result, it was shown that using our method, such simulations can be performed without errors or reduced accuracy, even if the number of rounds is set to the minimum value, i.e. 1. This implies that further speedup is possible. In our code, to check how much greater the computational cost of the random force is when compared to the computational cost for other conservative and dissipative forces, we calculated interaction forces for every particle pair regardless of whether they were inside or outside the cut-off radius. Finally, the generation of random numbers by the TEA with 8 rounds accounts for 40% of the total calculation time. Reducing the number of rounds cuts the random number generation cost to 1/8, yielding an overall simulation speed increase factor of 1.5. In the verification simulations, we fixed the number of particles allocated to each node. We make the assumption that by keeping the particle-to-node ratio constant we will be able to simulate larger systems without an increase in simulation time, simply by the addition of further nodes and the use of parallel computation. To test our method we examined whether the random numbers produced are distributed uniformly and to verify that there are no correlations between the random numbers. For the method which meets these two requirements, we

also perform further validation checks using the NIST test sets. Next, we compared temperature, velocity distributions, mean square displacements and radial distribution functions of the DPD system using our method to those obtained using one of the most popular random number generators, the Mersenne Twister [36]. We believe this constitutes a good benchmark for comparison as the Mersenne Twister is renown for its high degree of randomness and computational efficiency. However, the Mersenne Twister is not always suitable for use with distributed memory systems that do not share memory between nodes, and this is where the TEA method has an advantage.

To conclude our simulations, we also apply our method to a biomembrane system and perform a similar comparison. This is done in order to demonstrate the wider applicably of the proposed random number generation method beyond that of a pure water simulation.

In the results and discussion section, we review our conjecture that the quality of the random numbers produced via our method is sufficient and that a DPD-water system, as well as membrane systems, can be simulated accurately using TEA calculations with only a single round. This implies two things: the seed numbers we select for our DPD calculations already possess a high degree of randomness; and selecting highly random seed numbers will help us to generate random numbers which themselves have an even higher degree of randomness, such that they produce accurate results when used in DPD simulations. In this research, we propose an improvement to the random number generation method used in the DPD method in a parallel computing environment. This work will, therefore, form the basis for more efficient DPD simulations in the future.

## Methods

### Dissipative particle dynamics

In DPD simulations, groups of several molecules are represented by single-site particles. The time evolution of the interacting coarse-grained particles is governed by Newton's equation of motion. In our simulations particle masses are set to 1. The force is composed of three terms,

$$\boldsymbol{f}_i = \sum_{j(\neq i)} (\boldsymbol{F}_{ij}^{C} + \boldsymbol{F}_{ij}^{D} + \boldsymbol{F}_{ij}^{R}), \tag{1}$$

where $\boldsymbol{F}_{ij}^{C}$ is the conservative force, $\boldsymbol{F}_{ij}^{D}$ is the dissipative force and $\boldsymbol{F}_{ij}^{R}$ is the random force. The conservative force is expressed using a soft potential as follows,

$$\boldsymbol{F}_{ij}^{C} = \begin{cases} -a_{ij}\left(1 - \frac{r_{ij}}{r_c}\right)\boldsymbol{e}_{ij} & (r_{ij} \leq r_c) \\ 0 & (r_{ij} > r_c), \end{cases} \tag{2}$$

where $\boldsymbol{r}_{ij} = \boldsymbol{r}_j - \boldsymbol{r}_i$, $r_{ij} = |\boldsymbol{r}_{ij}|$ and $\boldsymbol{e}_{ij} = \boldsymbol{r}_{ij}/r_{ij}$. Here, $a_{ij}$ is a parameter which describes the maximum repulsion between particle $i$ and particle $j$, and $r_c$ is the cutoff distance. The dissipative and random forces are given by

$$\boldsymbol{F}_{ij}^{D} = \begin{cases} -\gamma \omega^{D}(r_{ij})(\boldsymbol{e}_{ij} \cdot \boldsymbol{v}_{ij})\boldsymbol{e}_{ij} & (r_{ij} \leq r_c) \\ 0 & (r_{ij} > r_c) \end{cases} \tag{3}$$

and

$$\boldsymbol{F}_{ij}^{R} = \begin{cases} \sigma \omega^{R}(r_{ij})\zeta_{ij}\Delta t^{-\frac{1}{2}}\boldsymbol{e}_{ij} & (r_{ij} \leq r_c) \\ 0 & (r_{ij} > r_c), \end{cases} \tag{4}$$

respectively, where $\boldsymbol{v}_{ij} = \boldsymbol{v}_j - \boldsymbol{v}_i$, $\gamma$ is the friction parameter, $\sigma$ is the noise parameter, $\Delta t$ is the size of a time step and $\omega^{\text{D}}$ and $\omega^{\text{R}}$ are $r$-dependant weight functions which we take as

$$\omega^{\text{D}}(r_{ij}) = [\omega^{\text{R}}(r_{ij})]^2 = \begin{cases} \left(1 - \frac{r_{ij}}{r_c}\right)^2 & (r_{ij} \leq r_c) \\ 0 & (r_{ij} > r_c). \end{cases} \tag{5}$$

Here, $\zeta_{ij}$ is a random variable with a Gaussian distribution. However, in DPD simulations, it has been found that uniformly distributed random variables can be also used [5]. In testing, we also can confirm that it is the case that both distribution types give the same results when used in DPD simulations. The dissipative force $\boldsymbol{F}_{ij}^{\text{D}}$ and the random force $\boldsymbol{F}_{ij}^{\text{R}}$ act as a thermostat in DPD simulations. Therefore the friction parameter and the noise parameter are connected by the fluctuation-dissipation theorem as follows

$$\sigma^2 = 2\gamma k_{\text{B}}T, \tag{6}$$

where $k_{\text{B}}$ is the Boltzmann constant and $T$ is the temperature.

In order to establish a baseline for comparisons for our DPD simulations, and to evaluate the performance of the modifications we make, we selected a simple test system. The details of which are discussed later in this paper.

## Tiny encryption algorithm

The tiny encryption algorithm (TEA) was developed by David Wheeler and Roger Needham [33]. The TEA is shown in Algorithm 1. $n$ is the number of rounds, $x_0$ and $x_1$ are plain text strings, which will be encrypted, the constant delta is a binary representation of the golden ratio, as specified by the original TEA, and k0, k1, k2 and k3 are the encryption keys, which we set as k0 = 3, k1 = 4, k2 = 5 and k3 = 6.

**Algorithm 1** The Tiny Encryption Algorithm

```
function TEA(n, x₀, x₁)
  sum = 0
  for n rounds do
    sum = sum + delta
    x₀ = x₀ + XOR(SHIFTLEFT(x₁, 4) + k0, SHIFTRIGHT(x₁, 5) + k1, x₁ +
sum)
    x₁ = x₁ + XOR(SHIFTLEFT(x₀, 4) + k2, SHIFTRIGHT(x₀, 5) + k3, x₀ +
sum)
  endfor
endfunction
```

It has been reported that the TEA is superior to other hash functions for the generation of random numbers in a parallel environment due to its low computational cost and high efficiency [31]. Today, it is widely used in DPD simulations which utilise parallel computation methods [24, 32]. The TEA is a Feistel type cipher that has a symmetric iterative structure. Within each round of the TEA, several bit operations like XOR, ADD and SHIFT are used. The quality of the random number produced by the algorithm depends on the number of rounds. Zafar et al. [31] suggested that 8 round is optimal when one considers the avalanche effect, which occurs after 6 rounds, and the results of NIST and DIEHARD tests. The TEA has been utilised in prior DPD simulations. Phillips et al. [32] also used 8 rounds, while Tang et al. [24] used 4 rounds, including preprocessing. In this paper, we check the accuracy of DPD simulations using two types of TEA for 1, 2, 3, 4, 5, 6, 8, 16 and 32 rounds. The first type is the unmodified original TEA, with two 32-bit inputs and two 32-bit outputs (Algorithm 2). In our DPD simulations, calculations are performed at 64-bit precision, so at the start of the TEA one

64-bit seed is split into two 32-bit numbers which are the inputs for the TEA. The products of the TEA are two new 32-bit random numbers, which are then subsequently combined to form one 64-bit random number. The name we give this type of TEA is 32TEA(n:number of rounds). We use the $x$ component of relative particle-particle separation distance, which is stored as a 64-bit number, as the 32TEA(n) seed number. The second type of TEA used in this simulation study has two 64-bit inputs and two 64-bit outputs (Algorithm 3). No tuning was performed for this algorithm and we use the same delta as was used in 32TEA(n). With this type of TEA, we do not need to combine numbers together and so we obtain two random numbers from each TEA calculation. The name we give to this type of TEA is 64TEA(n:number of rounds). We use the $x$ and $y$ components of relative particle-particle separation distances as the 64TEA(n) seeds numbers. It is not clear if 64TEA(n) will produce high-quality random numbers, as it has also not been tested. The concern with 64TEA(n) is that there are two input numbers and thus twice as many sources for possible correlation between successive number generation iterations. So we have included it in our comparison in order to see how DPD simulations are influenced by the input and output process of the algorithms used.

**Algorithm 2** Encryption Algorithm by 32TEA(*n*)

```
union Uni
  double SEED-NUMBER
  unsigned int seed-number[2]
  unsigned long long encrypted-number
end union
uni.SEED-NUMBER = distance
32TEAn(uni.seed-number[0], uni.seed-number[1])
rand = (double)uni.encrypted-number
```

**Algorithm 3** Encryption Algorithm by 64TEA(*n*)

```
union Uni
  double SEED-NUMBER[2]
  unsigned long long encrypted-number[2]
end union
uni.SEED-NUMBER[0] = distanceX
uni.SEED-NUMBER[1] = distanceY
64TEAn(uni.SEED-NUMBER[0], uni.SEED-NUMBER[1])
rand0 = (double)uni.encrypted-number[0]
rand1 = (double)uni.encrypted-number[1]
```

## Simulation conditions

**Water system.**   To increase computational efficiency when performing large-scale simulations, the system may be divided into multiple cells, using domain decomposition methods. If the shortest length of a cell is set to be slightly larger than that of the cutoff distance, then each node will perform calculations for a cell size of $1.5 \times 1.5 \times 1.5$ and it will be responsible for about 270 particles at a density of $\rho = 3$. Based on such considerations, we created systems containing 256 particles of coarse-grained water and performed DPD calculations with a single node. This also allows us to make a straightforward comparison with results obtained from DPD simulations which use the Mersenne Twister for random number generation. For the DPD parameters used in the current work, we referred to the paper by Groot and Warren [5]. Time evolution is calculated using a modified version of the velocity-Verlet algorithm with a DPD algorithm parameter $\lambda = 0.5$ and a time step $\Delta t = 0.04$. The $\gamma$ parameter, which is related to dissipative forces, was set to 6.75. The cut-off radius $r_c = 1.0$ and $k_B T = 1.0$. The random number generators used were 32TEA(1), 32TEA(2), 32TEA(3), 32TEA(4), 32TEA(6), 32TEA(8), 32TEA(16), 32TEA(32), 64TEA(1), 64TEA(2), 64TEA(3), 64TEA(4), 64TEA(6), 64TEA(8), 64TEA(16) and 64TEA(32). These generators were used together with the Box-Muller

method [37] to produce random numbers with a Gaussian distribution. For comparison, we also performed a series of separate simulations with uniformly distributed random numbers, for which we did not use the Box-Muller method. Our motivation for doing this was because Groot and Warren [5] reported that the results obtained from using uniformly distributed random numbers are not appreciably different from those which use Gaussian random numbers. Since the computational cost of uniform random number generation is lower, it should be seen as the better method of the two, if the results are the same.

The time required for random number generation was compared for each of the random number generation methods described above. Although the TEA method requires over 8 rounds to generate high-quality random numbers, we briefly checked two properties of these random numbers. First, we examined whether the random numbers were uniformly distributed in the set interval. Second, we performed t-tests [38] to investigate whether there were any correlations between successively generated random numbers. This check is necessary since we know that there are correlations between the seed numbers, i.e. distances of particle pairs in consecutive steps. We then performed additional checks to measure the randomness of the generated numbers using the NIST test sets. Also, in order to confirm whether the random force and the dissipation force work well as a thermostat, the velocity distribution of the particles and temperature were investigated. Furthermore, the mean square displacement and the radial distribution function were examined to confirm the dynamic and static properties of the system.

**Membrane system.** To further demonstrate the validity of our approach, we tested the applicability of this method to a more complicated system. The system consists of 108,000 particles, 17% of which are POPC lipids and the rest are water particles. Each parameter was chosen to match those of a previous study [39]. These simulations were performed using parallel computing with 32TEA(1) and 32TEA(32). A comparison between the results obtained using these two algorithms is presented in the next section, along with our results from the previously described pure water simulations.

## Results and discussion

### Water system

To know the time required for the random number generation process, we generated many random numbers ($1.0 \times 10^{11}$) and measured the time as shown in Fig 1. As expected, it was found that the calculation cost is lower for a lower number of rounds for both 32TEA and 64TEA. In the paper of Zafar et al., [31] it was observed that as the number of rounds decreases in 32TEA, the quality of the generated random numbers also becomes worse. Although We already know that the properties of the generated random numbers are not as good as they could be, we have evaluated the properties we expect to be the minimum necessary for use in DPD simulation as follows.

First, we confirmed whether or not the generated random numbers are uniformly distributed. In one time-step, there will be close to $(256 \times 256 - 256)/2 = 32640$ pairs in the cell, and so, for the purposes of this test, we generated this many random numbers. We defined a null hypothesis as "the 32640 randomly generated numbers be distributed uniformly within in the section [0, 1]". Then, we performed a $\chi^2$-test [40] for these random numbers to determine whether the hypothesis is valid or not. In this case, we divided the section [0, 1] into 16 subsections and counted the number of random numbers which appeared in each subsection. If the random numbers are indeed uniformly distributed, then approximately 2040 random numbers will appear in each subsection. A measure of uniformity can then be estimated by

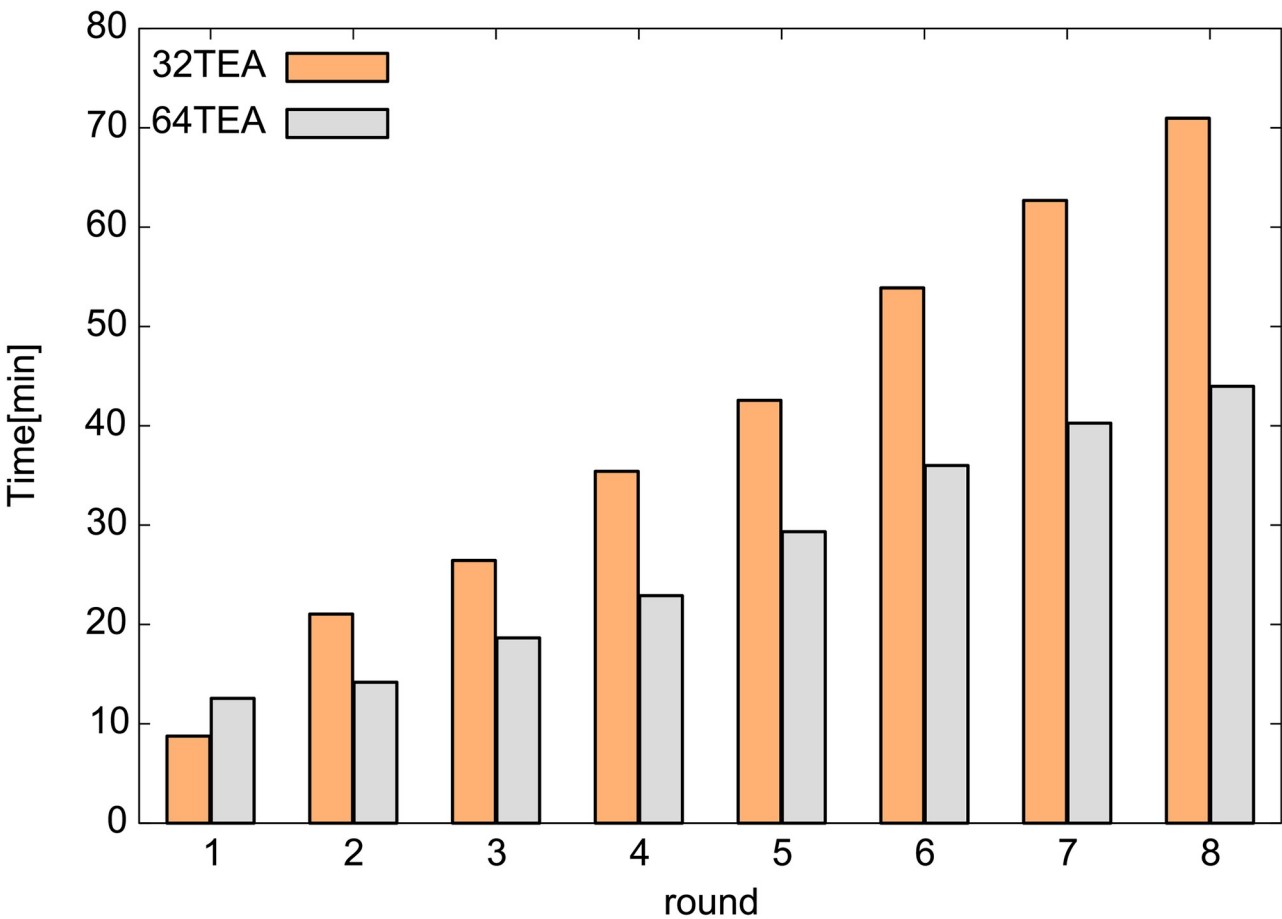

**Fig 1. The required time to generate $1.0 \times 10^{11}$ random numbers.**

calculating $A$:

$$A = \sum_{i=1}^{n} \frac{(n_i - N)^2}{N}, \tag{7}$$

where $n$ is the number of subsections which follows Sturges' rule [41] i.e. 16, $n_i$ is the number of random numbers generated in the $i$th subsection, and $N$ is the theoretical expected frequency i.e. 2040, as is explained above. It is known that $A$ follows a $\chi^2$ distribution with $n-1$ degrees of freedom. In order to test the null hypothesis, we may, therefore, compare $A$ with the value corresponding to a significance level of 1%.

Second, the relative particle-particle separation distances, which are used for TEA seeds, are correlated because DPD particle positions are strongly dependent on the positions in the previous time-step, so, in our t-tests we observed a correlation between the random numbers generated in consecutive calculation steps, for specific particle-pairs. For this test, we used 9500 random numbers which were taken from a total of 10000 steps, excluding the first 500, and defined a null hypothesis for the t-test as "the 9500 random numbers generated in consecutive calculation steps are not correlated". In our case, to check for sequential correlations in the

series of 9500 numbers ($x_i$: $1 \leq i \leq 9500$), we calculated the correlation coefficient $r$:

$$r = \frac{\frac{1}{n}\sum_{i=1}^{n} x_i x_{i+1} - \bar{x}}{\frac{1}{n}\sum_{i=1}^{n}(x_i - \bar{x})^2}, \tag{8}$$

where n is the number of random numbers, i.e. 9500, $\bar{x} = \frac{1}{n}\sum_{i=1}^{n} x_i$. We then derived $B$ using $r$ from the following expression:

$$B = \sqrt{n-2}\,\frac{r}{\sqrt{1-r^2}}. \tag{9}$$

It is known that $B$ follows a $t$ distribution with $n-2$ degrees of freedom. As before, to consider the null hypothesis, we compared the calculated $B$ values with the value corresponding to a significance level of 1%.

In Fig 2 we present results of the $\chi^2$-test and t-test, respectively. The red lines in these plots indicate a significance level of 1%. The 64TEA method generates two random numbers per use, the second of which is expected to be of slightly higher quality, so these were plotted separately as 64TEA(n)a and 64TEA(n)b. As can be seen from Fig 2(a), for 64TEA(n), three, or more than three, rounds of 64TEA can generate random numbers that are sufficiently uniform in their distribution. For 32TEA only one round is required to meet this standard. In the t-test, unlike with the $\chi^2$-test, only one of the two random numbers of 64TEA with two rounds can pass this test. In the case of 32TEA, it was again seen that a single round is sufficient to ensure that the correlations are below a 1% significance level. In Fig 2, only the minimum number of rounds required to pass each test is shown because we have confirmed that the same requirements are satisfied when using a higher number of rounds.

As shown in Figs 1 and 2, when we compare the computational cost and the quality of random numbers produced, it appears that 32TEA(1) is the optimal algorithm at this stage of our analysis. To confirm that this is indeed that case, we also tested the randomness of the generated numbers using the NIST test sets for 32TEA. The NIST random number test sets consist of 15 different test methods. Each test method is performed for each series of 1048576 bits, and the results of the 1000 series are used to make a comprehensive judgment. Since these are statistical tests of a random stream, the result is provided in the form of a p-value for each test or subtest, indicating the probability that the result is due to random chance. We use a significance level of 0.01 and were able to confirm that 32TEA(1) passed all tests. We were also able to confirm that 32TEA(2), 32TEA(3), 32TEA(4), 282 32TEA(5), 32TEA(6), 32TEA(7), 32TEA(8) and 32TEA(32) pass every NIST test. In a previous study, it was shown that the conventional implementation of the 32TEA(1) was not sufficient to pass NIST tests [31]. However, by using the relative positions of particle pairs as seed numbers, we can generate random numbers with sufficiently high randomness when using only one round.

For the water system simulated using Gaussian random numbers, the temperature was measured from the kinetic energy of the system in order to confirm whether the temperature was correctly controlled or not (Fig 3(a)). To further verify this, we also calculated the mean temperature and its fluctuations from the final 9500 steps out of a total of 10000 steps (Fig 3(b)). From our results, we can confirm that the temperature was accurately controlled when the system was simulated with a TEA which passes the $\chi^2$-test and t-test detailed above. This is the case even when not using conventional TEAs. Furthermore, in Fig 4, we show a

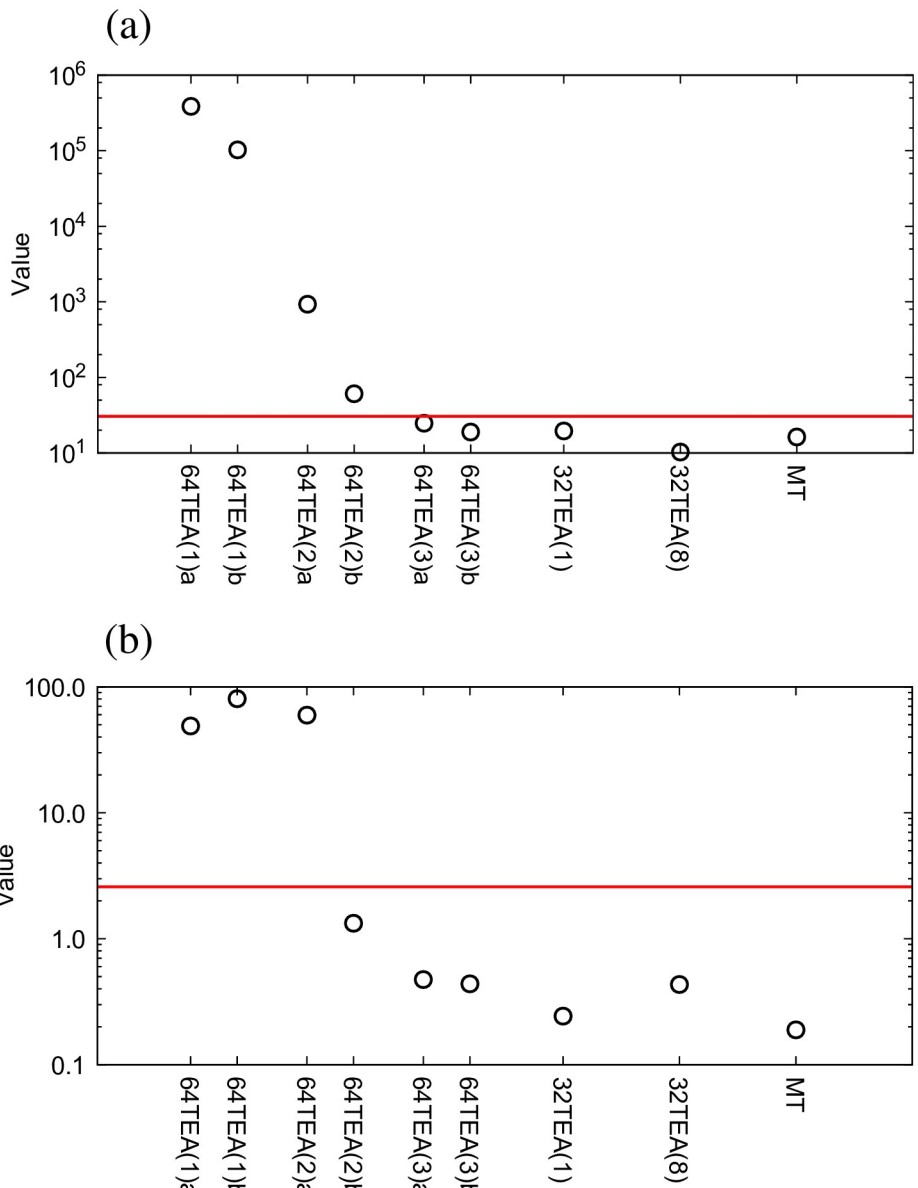

**Fig 2.** The results of (a) $\chi^2$-test, (b) t-test. The red line indicates a significance level of 1%.

comparison of the velocity distribution with theoretically predicted values and error rates. The error rate was calculated as follows,

$$P(v_{\text{error}}) = \frac{|P(v_i) - P(v_{\text{theoretical}})|}{P(v_{\text{theoretical}})}, \tag{10}$$

where $P(v_i)$ is the probability distribution function of $v_i (i = x, y, z)$, and $P(v_{\text{theoretical}})$ is its theoretically calculated counterpart. From this figure, it seems that the simulated distribution is a close match to the theoretical one. The median value ($v = 0$) has an error of about 2%, which is the same error rate as found with the Mersenne Twister. (Fig 4(c)) On the other hand, for the TEA versions which did not pass the above tests, such as 64TEA(2), we may see from

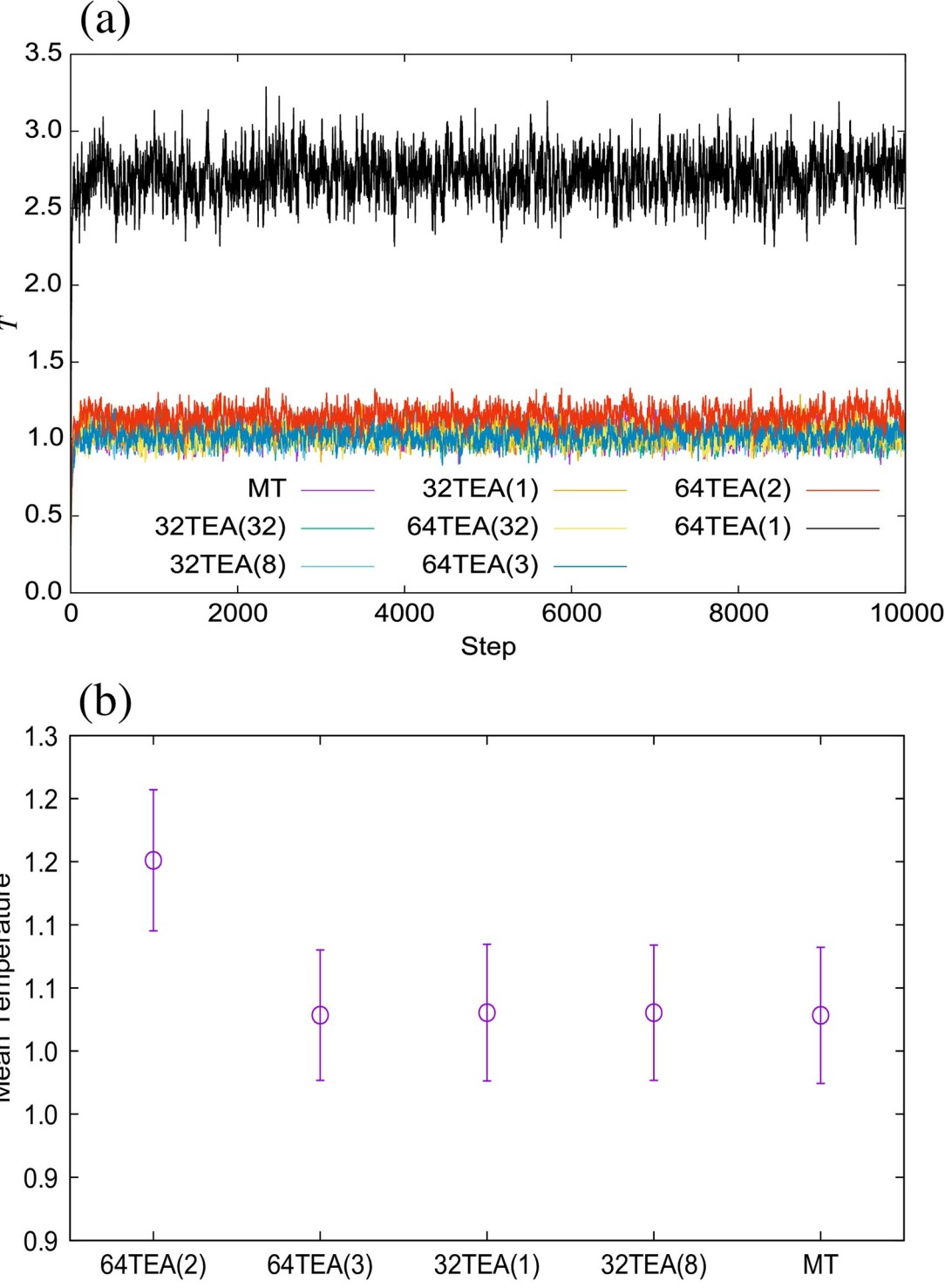

**Fig 3.** (a) Temperature of the water system, (b) mean temperature and its fluctuations.

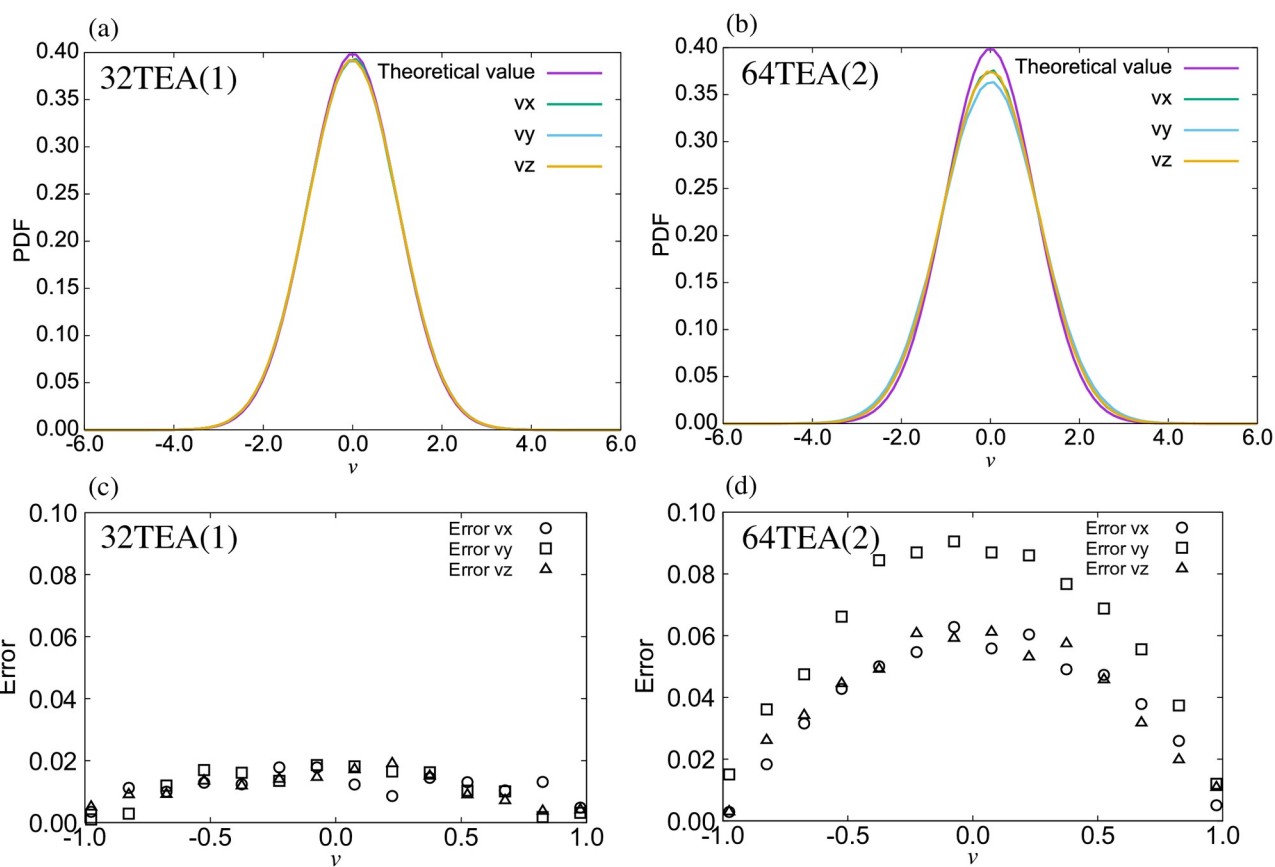

**Fig 4.** Probability distribution function (PDF) of the velocity for the water system using (a) 32TEA(1) and (b) 64TEA(2) and the error rate of the velocity distributions using (c) 32TEA(1) and (d) 64TEA(2).

Fig 4(b) and 4(d) that there are discrepancies with the theoretical values, and that there are inconsistencies in the different cartesian directions for the velocities. In some cases, we found differences in the location of the center of these histograms, and in other cases, we found differences in the height of the highest point of the histograms.

Next, we investigated the dynamic properties of the DPD method for each type of TEA. The mean square distribution (MSD) is shown in Fig 5(a). We also calculated the error rate of the diffusion coefficient (Fig 5(b)) from the following expression,

$$D_{\text{error}} = \frac{|D(\text{TEA}) - D(\text{MT})|}{D(\text{MT})} \tag{11}$$

where $D(\text{TEA})$ is the diffusion coefficient of the water system when using a given TEA, and $D(\text{MT})$ is the diffusion coefficient of the water system obtained through use of the Mersenne twister. From these figures, we observe that the difference between the water system simulated with 32TEA(1) and those with Mersenne twister is lower than it is for 32TEA(8), which is the more commonly used method. The same is true of static properties. The radial distribution function (RDF) is shown in Fig 6. From this picture, it is clear that the RDF of the system with 64TEA(1) is considerably different from those with other algorithms, especially around $r = 0$. This demonstrates the effect of strong correlations between seeds numbers, which in this case are relative particle-particle separation distances. The random number algorithms cannot

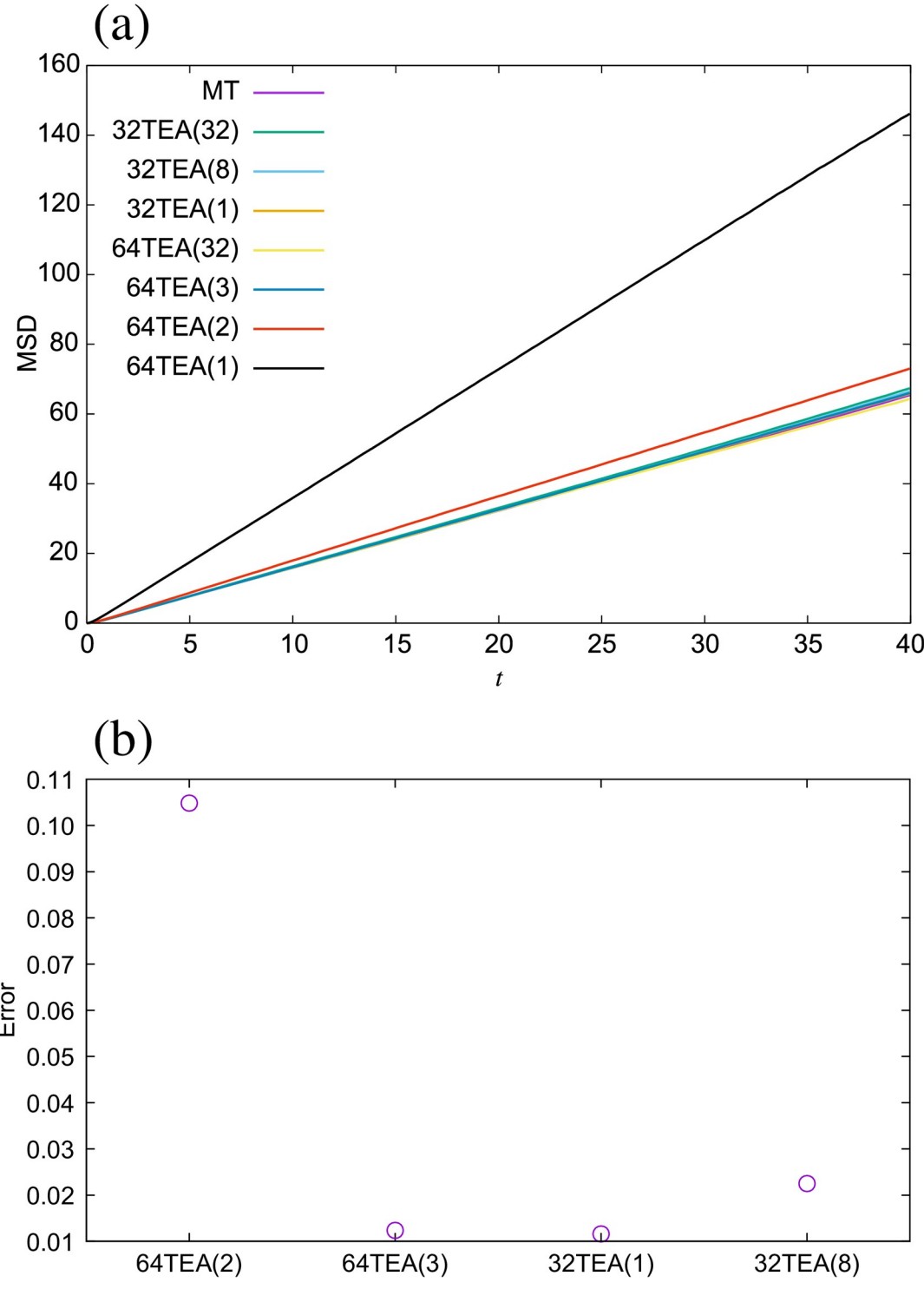

**Fig 5.** (a) Mean square displacement for the water system, (b) error rate of the diffusion coefficient.

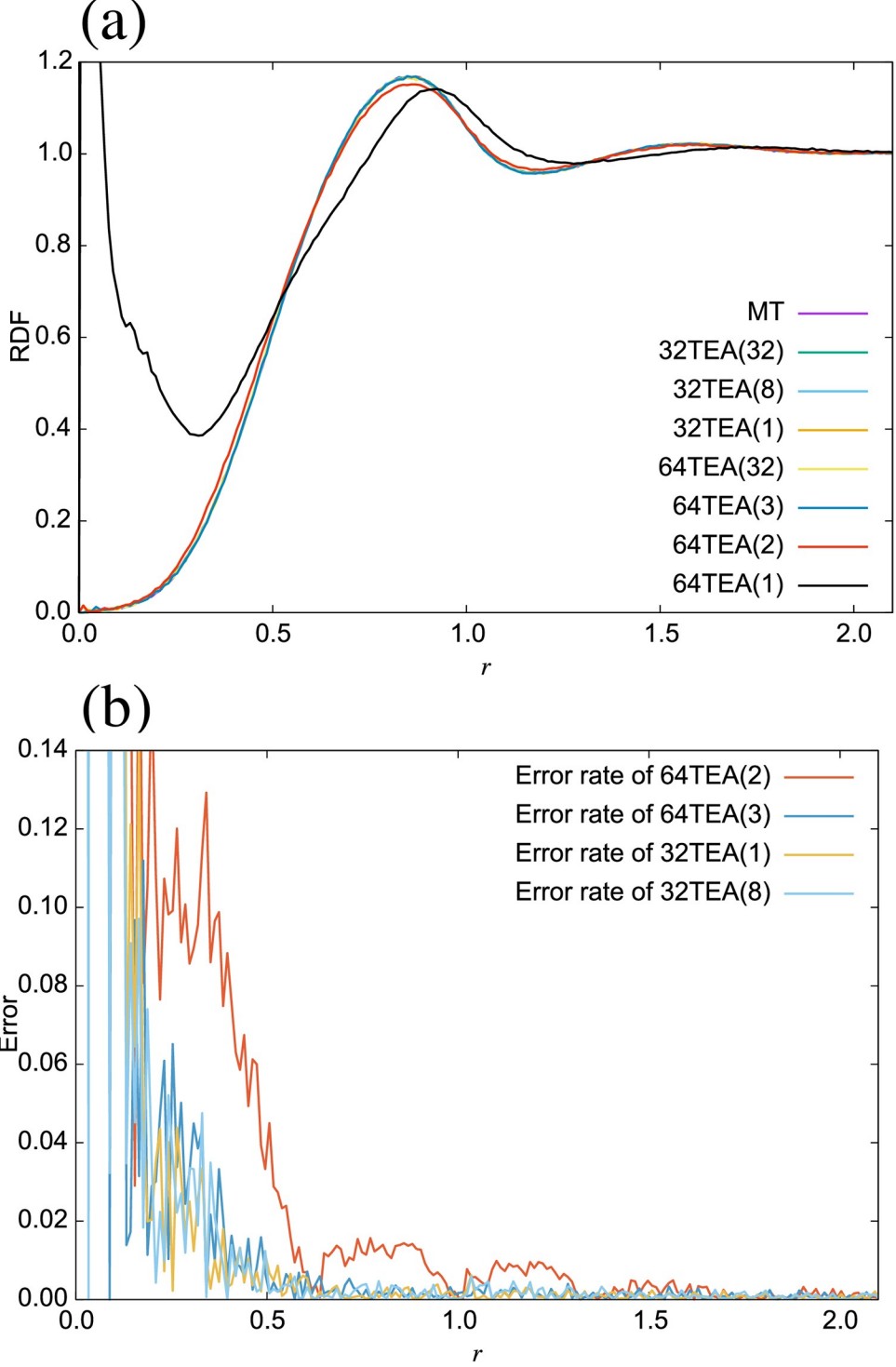

**Fig 6.** (a) Radial distribution function, (b) error rate of radial distribution function.

completely remove these correlations. In this case, for small seed numbers, which correspond to two particles that are in close proximity, some specific values are generated, which eventually result in negative random forces, and the particle pairs may overlap. This point serves to highlight the importance of ensuring minimal correlation between successively generated random numbers. The error rate of the RDF (Fig 6(b)) is calculated as below,

$$RDF_{\mathrm{error}} = \frac{|RDF(r)_{\mathrm{TEA}} - RDF(r)_{\mathrm{MT}}|}{RDF(r)_{\mathrm{MT}}} \tag{12}$$

We find that only 64TEA(1) and 64TEA(2) has a periodic error in its RDF.

Moreover, we also compared the water system which used uniform random numbers generated from 32TEA(1) and the water system with uniform random numbers generated using the Mersenne twister. Using the same analysis described above, we find that using 32TEA(1) gives an acceptable level of accuracy, which is in agreement with the findings of prior work in this area [5] (See S1 File).

For the 32TEA presented here, our evaluation using the NIST test sets confirms that it passes these tests. Therefore, we can conclude that the 32TEA(1) can be used to generate random numbers with a high degree of randomness.

## Membrane system

In this section, we describe the reconstruction of a biomembrane structure using POPC lipids. This type of biomembrane has been studied in detail in previous studies and we follow the same approach here [39]. The temperature (Fig 7), pressure in each direction (Fig 8), and thickness of the biomembrane were determined from the density profiles of each particle (Fig 9). No difference was found between the simulation results for 32TEA(1) and 32TEA

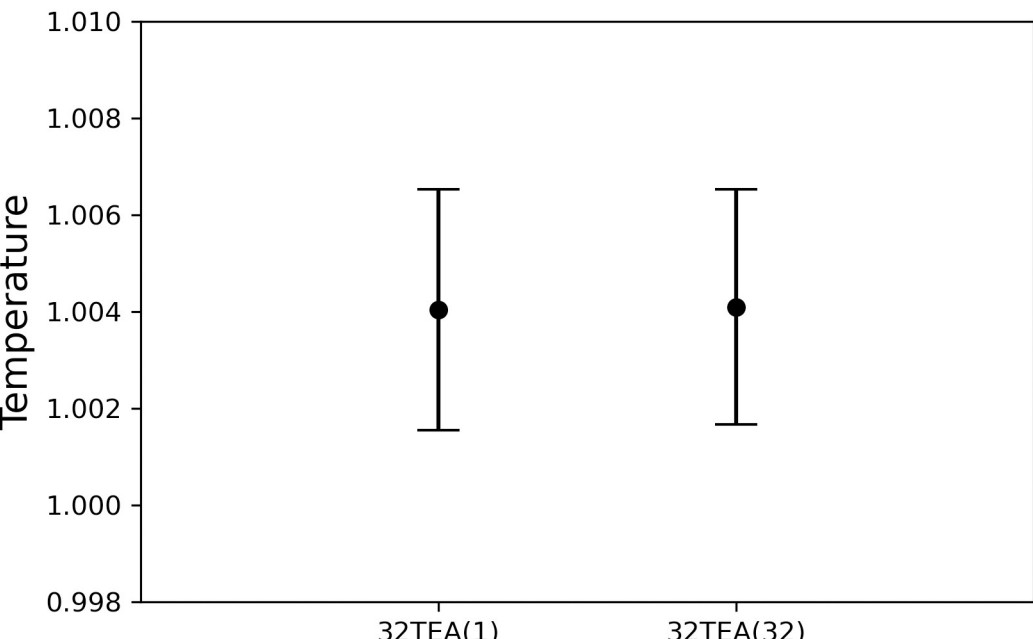

**Fig 7. Temperature of the membrane system.**

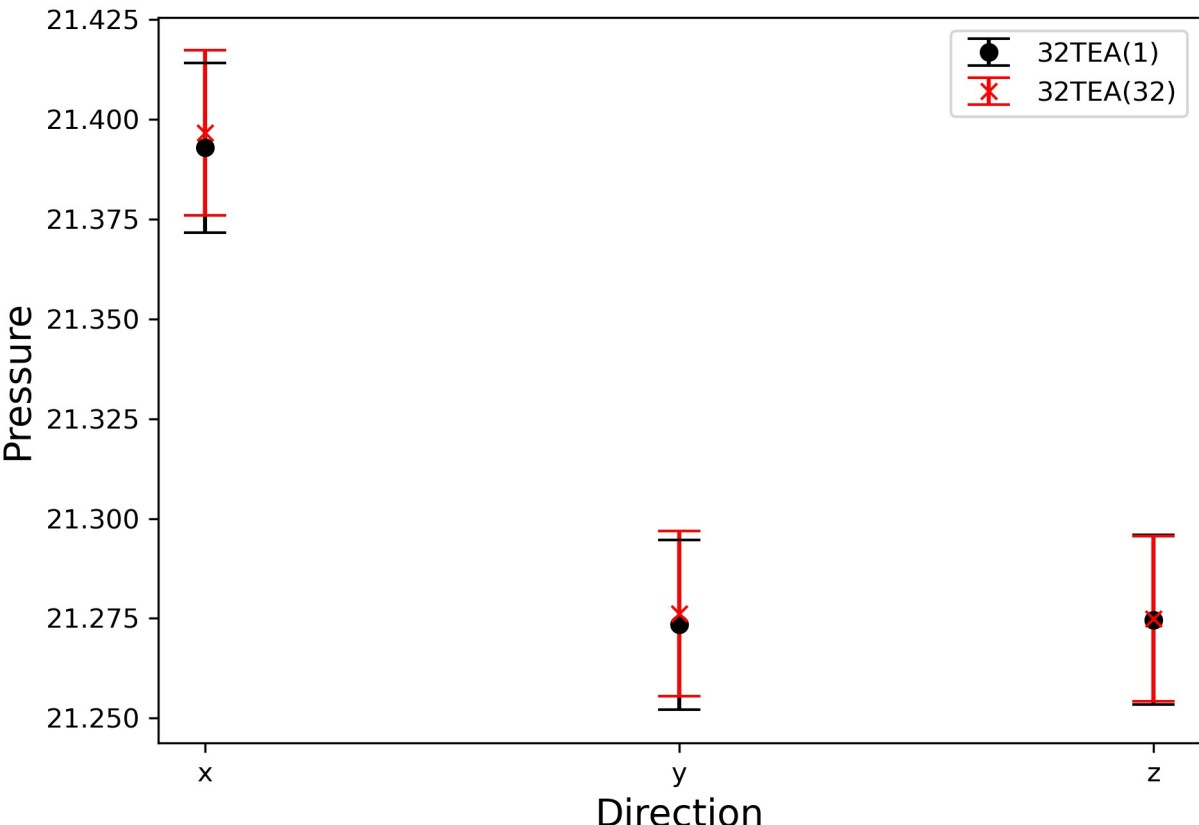

**Fig 8. Pressure of the membrane system for each direction.**

(32). We can therefore say that, when using the distance between particles is used as the seed number, only one round function of TEA is sufficient to produce accurate simulation results.

## Conclusion

In this research, we tried to reduce the calculation cost of random number generation when using TEA algorithms and to check whether the properties of these random numbers will impact the results of DPD simulations. From these results, it has been shown that when using 64-bit particle-particle separation distances as 32TEA(1) seeds, random numbers of sufficient quality can be generated for use in DPD simulations. Usually, the TEA does not perform well enough with only one round, which includes only a few bit operations. However, these results suggest the possibility that simple bit processing is sufficient for DPD simulations, which benefit from the randomness in seed derived from DPD calculation itself. A good example of such seed numbers may be found in the form of relative particle-particle separation distances in molecular simulations. Taking these findings into consideration, the cost of random number generation via this new method is one-eighth that of the most commonly used alternative, where 8 rounds are required, and our DPD code achieved a speed-up factor of 1.5. Further to this, as reported by Groot and Warren, [5] the same statistical results may be achieved without Gaussian random numbers, using instead uniform random numbers generated by 32TEA(1). This paper presents the possibility of accelerating DPD calculations on distributed memory

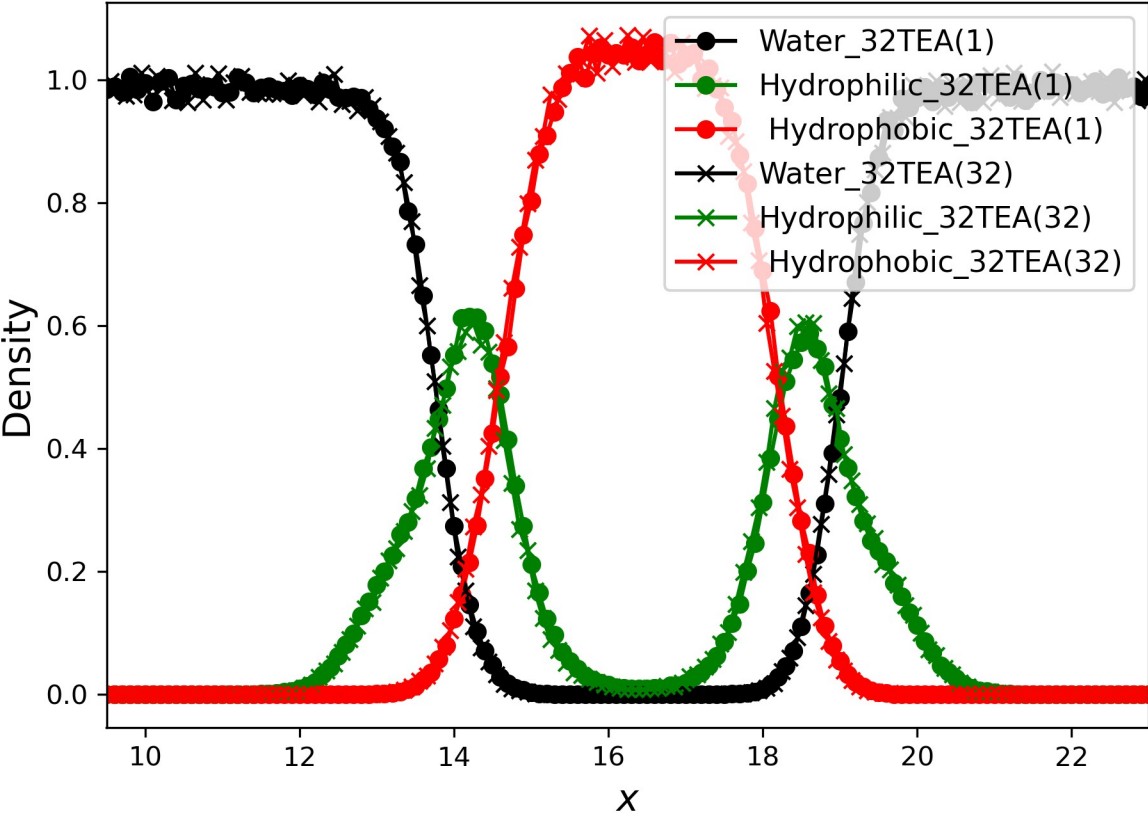

**Fig 9. Density plot of each particle in the *x* direction.**

systems and gives a detailed review of the use of various hash functions as random number generators.

## Supporting information

**S1 File.**
(PDF)

## Author Contributions

**Conceptualization:** Kiyoshiro Okada, Kenji Yasuoka.

**Data curation:** Kiyoshiro Okada.

**Formal analysis:** Kiyoshiro Okada.

**Investigation:** Kiyoshiro Okada.

**Methodology:** Kiyoshiro Okada, Paul E. Brumby, Kenji Yasuoka.

**Project administration:** Kenji Yasuoka.

**Supervision:** Kenji Yasuoka.

**Validation:** Kiyoshiro Okada.

**Writing – original draft:** Kiyoshiro Okada.

**Writing – review & editing:** Kiyoshiro Okada, Paul E. Brumby, Kenji Yasuoka.

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
