## [Decision Letter · Decision Letter 0]

8 Feb 2021

PONE-D-21-01494

The influence of random number generation in dissipative particle dynamics simulations using a cryptographic hash function

PLOS ONE

Dear Dr. Yasuoka,

Thank you for submitting your manuscript to PLOS ONE. After careful consideration, we feel that it has merit but does not fully meet PLOS ONE’s publication criteria as it currently stands. Therefore, we invite you to submit a revised version of the manuscript that addresses the points raised during the review process.

As you will see from the reports, both reviewers consider your study as fundamental and practically useful. There are several technical questions which I hope you will be able to answer. One of the reviewers suggests language improvement and shortening standard description of the DPD procedure. He/she is also concerned with low significance of the results relative to other publications on the DPD methodology.  Please think how to strengthen the importance of your contribution, may be the discussion on practical implementation of your results could be extended and some new examples considered. Generally, you need to convince readers that by using your approach they will really gain something from it. 

We look forward to receiving your revised manuscript.

Kind regards,

Yaroslav V. Kudryavtsev, Ph.D.

Academic Editor

PLOS ONE

Journal Requirements:

Reviewers' comments:

Reviewer's Responses to Questions

**Comments to the Author**

1. Is the manuscript technically sound, and do the data support the conclusions?

Reviewer #1: Yes

Reviewer #2: Partly

2. Has the statistical analysis been performed appropriately and rigorously? 

Reviewer #1: Yes

Reviewer #2: Yes

3. Have the authors made all data underlying the findings in their manuscript fully available?

Reviewer #1: Yes

Reviewer #2: Yes

4. Is the manuscript presented in an intelligible fashion and written in standard English?

Reviewer #1: Yes

Reviewer #2: No

5. Review Comments to the Author

Reviewer #1: In this study, Okada et al investigate some modifications to the tiny encryption algorithm (TEA) that has been used for generating uniformly distributed random numbers for parallel dissipative particle dynamics simulation codes. The modifications include 1) varying the number of rounds for encrypting the inputs (up to 32), 2) varying the precision of the inputs and outputs (32-bit and 64-bit per input channel) and 3) using the x or (x and y) components of the particle-particle separation vector as inputs. The authors examine the quality of the modified algorithms by performing statistical tests for the randomness of the generated numbers, i.e their uniformity and the correlation between successively generated random numbers. The algorithms are then used in simulations of coarse-grained water, to see how they influence the mean squared displacement (MSD), radial distribution function (RDF) and temperature of the water particles. According to their tests, they found that the 32TEA(1) algorithm gives the lowest computational cost, comparable quality to the Mersenne Twister algorithm, and highly accurate results regarding the controlled temperature, structural (RDF) and dynamic (MSD) results.

Overall I think this is a useful and quite carefully performed study, and recommend its publication in PLOS ONE. There are a couple of items I suggest the authors consider.

1) For the modified pseudo-RNGs to be adopted for other systems than the test case, the authors need more rigorous tests than the current tests on a rather small set of generated random numbers (9500), which is well below the prime period of the RNGs. See for example Ref. 45 for the commonly used tests for PRNGs.

2) The finding that 32TEA(1) is the optimal algorithm in terms of randomness is indeed interesting.

Can the authors attempt to explain why this might be the case, otherwise it is difficult to envision if the 32TEA(1) would work as well for other use cases beyond the test random number set and the particular simulation composed of 270 coarse-grained water molecules at liquid state. Also in Fig 2, for 32TEA algorithms only data for n=1 and n=8 are shown. What are the results for other values of n for 32TEA?

3) A related study for implementing and comparing the effects of different PRNGs including TEA(8) for parallel DPD simulations (along the line with Phillips el al (Ref 43), and Afshar et al (ref 45)) where more quantities and systems, that are typical of DPD studies, were considered is Nguyen and Plimpton, Computational Materials Science 2015, 100, 173-180.

Reviewer #2: This paper describes a fundamental study on the influence of the quality of the random number generation in dissipative particle dynamics (DPD). They show that a reduced number of rounds in the generation of random numbers could still produce accurate DPD simulations. I find the topic of this work is interesting; however, compared to previous simulation studies, its contribution is obviously insignificant.

Other points:

1. Page 1: DPD simulations allow for the use of longer time-steps when compared with many other CG approaches, such as the MARTINI force-field. Here, you mentioned that MARTINI force field, rather than an approach. Hence, it is an inaccurate statement.

2. Pages 3-4: There is no need to discuss DPD fundamentals in details. It has been done in a lot of publications.

6. PLOS authors have the option to publish the peer review history of their article (what does this mean?). If published, this will include your full peer review and any attached files.

Reviewer #1: No

Reviewer #2: No

---

## [Author Response · Author response to Decision Letter 0]

8 Apr 2021

I have upload our response to specific reviewer and editor comments as "Response to Reviewers" in Attach Files section.

---

## [Editor Report · Decision Letter 1]

12 Apr 2021

The influence of random number generation in dissipative particle dynamics simulations using a cryptographic hash function

PONE-D-21-01494R1

Dear Dr. Yasuoka,

We’re pleased to inform you that your manuscript has been judged scientifically suitable for publication and will be formally accepted for publication once it meets all outstanding technical requirements.

Kind regards,

Yaroslav V. Kudryavtsev, Ph.D.

Academic Editor

PLOS ONE

Additional Editor Comments (optional):

The authors have taken into account the critical comments from the reviewers, introduced some new material, and reorganized the text in the proper way. I believe that after revision the paper meets the criteria for publication in PLOS One and does not require further scientific review.
---

## [Editor Report · Acceptance letter]

16 Apr 2021

PONE-D-21-01494R1 

The influence of random number generation in dissipative particle dynamics simulations using a cryptographic hash function  

Dear Dr. Yasuoka:

I'm pleased to inform you that your manuscript has been deemed suitable for publication in PLOS ONE. Congratulations! Your manuscript is now with our production department. 

Kind regards, 

on behalf of

Dr. Yaroslav V. Kudryavtsev 

Academic Editor

PLOS ONE